# Towards a Theory of AI Personhood

## Abstract

I am a person and so are you. Philosophically and legally, we sometimes grant personhood to non-human animals, and even to entities such as rivers and corporations. But when, if ever, should we ascribe personhood to AI systems? In this paper, we outline necessary conditions for AI personhood, focusing on *agency*, *theory-of-mind*, and *self-awareness*. We discuss evidence from the machine learning literature regarding the extent to which contemporary AI systems, such as language models, satisfy these conditions. We argue that no current AI system could plausibly be considered a person.

## 1 Introduction

Contemporary AI systems are built "in our image". They are trained on human-generated data to display person-like characteristics, and are easily anthropomorphised (Shanahan et al., 2023). These systems are already being incorporated into everyday life as generalist assistants, "friends", and even artificial romantic partners (OpenAI, 2024b; Pierce, 2024; Depounti et al., 2023). In the coming years, AI systems will continue to become more capable, and more integrated into human society.

Taking technological trends, and the accompanying philosophical questions, seriously, Russell asks "What if we succeed?" (Russell, 2019). Russell's answer is a focus on the problem of how to *control* AI agents surpassing human capabilities. Accordingly, there is growing literature on the problem of aligning AI systems to human values (Ngo et al., 2024; Bales et al., 2024; Gabriel, 2020; Christian, 2021).

Beyond this, there are broader philosophical questions regarding whether AI systems can be ascribed properties like belief (Herrmann and Levinstein, 2024), intent (Shanahan et al., 2023; Ward et al., 2024), agency (Kenton et al., 2022), theory-of-mind (Strachan et al., 2024), self-awareness (Laine et al., 2024), and even consciousness (Butlin et al., 2023; Shanahan, 2024; Seth, 2024).

It is thus timely to start considering a future society in which humans share the world with AI systems possessing some, or all, of these properties. Future AI systems may have claims to moral or political status (Ladak, 2024; Sebo and Long, 2023), but, because their natures differ in important respects from those of human beings, it may not be appropriate to simply apply existing norms in the context of AI (Bostrom and Shulman, 2022). Although these considerations may seem like science fiction, fiction reflects our folk intuitions (Rennick, 2021), and sometimes, life imitates art.

As humans, we already share the world with other intelligent entities – such as animals, corporations, and sovereign states. Philosophically and legally, we often grant *personhood* to these entities, enabling us to harmoniously co-exist with agents that are either much less, or much more, powerful than individual humans (Martin, 2009; Group, 2024).

This paper advances a theory of AI personhood. Whilst there is no philosophical consensus on what constitutes a person (Olson, 2023), there are widely accepted themes which, we argue, can be practicably applied in the context of AI. Briefly stated, these are 1) agency, 2) theory-of-mind (ToM), and 3) self-awareness. We explicate these themes in relation to technical work on contemporary systems.

## 2 Conditions of AI Personhood

When should we ascribe *personhood* to AI systems? Building on Dennett (1988); Frankfurt (2018); Locke (1847), and others we outline three core conditions for AI personhood.

Submitted to 38th Conference on Neural Information Processing Systems (NeurIPS 2024). Do not distribute.

**Agency.** Persons are entities with mental states, such as beliefs, intentions, and goals (Dennett, 1988; Strawson, 2002; Ayer, 1963). In fact, there are many entities which are not persons but which we typically describe in terms of beliefs, goals, etc (Frankfurt, 2018), such as non-human animals, and, in some cases, either rightly or wrongly, AI systems. Dennett calls this wider class of entities *intentional systems* – systems whose behaviour can be explained or predicted by ascribing mental states to them (Dennett, 1971).

In the context of AI, such systems are often referred to as *agents* (Kenton et al., 2022).The standard philosophical theory says that agency is the capacity for *intentional action* – action that is caused by an agent's mental states, such as beliefs and intentions (Schlosser, 2019). Similar to Dennett, our first condition for AI personhood is *agency* (Dennett, 1988).

Many areas of AI research focus on building *agents* (Wooldridge and Jennings, 1995). Formal characterisations often focus on the *goal-directed* and *adaptive* nature of agency. For instance, economic and game-theoretic models focus on *rational* agents which *choose actions to maximise utility* (Russell and Norvig, 2016). Belief-desire-intention models represent the agent's states explicitly, so that it selects intentions, based on its beliefs, in order to satisfy its desires (Georgeff et al., 1999). Reinforcement learning (RL) agents are trained with feedback given by a reward function representing a goal and learn to adapt their behaviour accordingly – though, importantly, the resultant agent may not internalise this reward function as *its goal* (Shah et al., 2022; Turner, 2022). Wooldridge and Jennings; Kenton et al.; Shimi et al. provide richer surveys of agency and goal-directedness in AI.

When should we describe artificial agents as *agents* in the philosophical sense? The question of whether AI systems "really have mental states" is contentious, and anthropomorphic language can mislead us about the nature of systems which merely display human-like characteristics (Shanahan et al., 2023). However, a range of philosophical views would ascribe beliefs and intentions to certain AI systems. For example, dispositionalist theories determine whether an AI system believes or intends something, depending on how it's disposed to act (Schwitzgebel, 2024a; Ward et al., 2024). Under another view, representationalists might say an AI believes $p$ if it has certain internal representations of $p$ (Herrmann and Levinstein, 2024). Furthermore, we can take the "intentional stance" towards these systems to apply terms like belief and goals, just when this is a *useful description* (Dennett, 1971). Indeed, Kenton et al. (2022) take the intentional stance to formally characterise agents as systems which adapt their behaviour to achieve goals.

Given the substantial philosophical uncertainty regarding how we might determine whether AI systems have mental states, adopting the intentional stance enables us to describe these systems in intuitive terms, and to precisely characterise their behaviour, without exaggerated philosophical claims. Hence, we can describe AI systems as *agents* to the extent that they adapt their actions *as if* they have mental states like beliefs and goals.

Certain narrow systems, such as RL agents, might adapt to achieve their goals in limited environments (for example, to play chess or Go), but may not have the capacity to act coherently in more general environments. In contrast, relatively general systems, like LMs, may adapt for seemingly arbitrary reasons, such as spurious features in the prompt (Sclar et al., 2024). We might be more inclined to ascribe agency to systems which adapt robustly across a range of general environments to achieve coherent goals. Such robust adaptability suggests that the system has internalised a rich causal model of the world (Richens and Everitt, 2024), making it more plausible to describe the system as possessing beliefs, intentions, and goals (Ward et al., 2024; MacDermott et al., 2024; Kenton et al., 2022).Hence, our first condition can be captured by the two following statements.

**Condition 1: Agency.** An AI system has *agency* to the extent that

    1. It is useful to describe the system in terms of mental states such as beliefs and goals.

    2. It adapts its behaviour robustly, in a range of general environments, to achieve coherent goals.

To what extent do contemporary LMs have agency? Many researchers are sceptical that LMs could be ascribed mental states, even in principle (Shanahan et al., 2023; Bender et al., 2021). On the other hand, much work has focused on trying to infer things like belief (Herrmann and Levinstein, 2024), intention (Ward et al., 2024), causal understanding (Richens and Everitt, 2024), spatial and temporal reasoning (Gurnee and Tegmark, 2024), general reasoning (Huang and Chang, 2023), and in-context learning (Olsson et al., 2022) from LM internals and behaviour. Many of these properties seem to emerge in large-scale models (Wei et al., 2022) and frontier systems like GPT-4 exhibit human-level performance on a wide range of general tasks (Chowdhery et al., 2023; Bubeck et al., 2023).

Do contemporary LMs have goals? LMs are typically pre-trained for next-token prediction and then fine-tuned with RL to act in accordance with human preferences (Bai et al., 2022). RL arguably increases LMs' ability to exhibit coherently goal-directed behaviour (Perez et al., 2022). Furthermore, LMs can be incorporated into broader software systems (known as "LM agents") which equip them with tools and affordances, such as internet search (Xi et al., 2023; Davidson et al., 2023). RL fine-tuning can enable LM agents to effectively pursue goals over longer time-horizons in the real world (OpenAI, 2024a; Schick et al., 2023).

**Theory-of-Mind.** Agents possess beliefs about the world, and within this world, they encounter other agents. An important part of being a person is recognising and treating others as persons. This is expressed in the philosophies of Kant; Dennett; Buber; Goffman et al.; Rawls and others. Kant, for instance, states that rational moral action must never treat other persons as merely a means to an end.

Treating others as persons necessitates understanding them as such – in Dennett's terms, it involves *reciprocating* a stance. Hence, in addition to having mental states themselves, AI persons should understand others by ascribing mental states to them. In other words, AI persons should have a capacity for *theory-of-mind (ToM)*, characterised by higher-order intentional states (Frith and Frith, 2005), such as beliefs about beliefs, or, in the case of deception, intentions to cause false beliefs (Mahon, 2016).

Language development is an indicator of ToM in children (Bruner, 1981). It's plausible that some animals have a degree of ToM. However, it's less plausible that any non-human animals have the capacity for *language*, excluding them, in some views, from being persons (Dennett, 1988). But LMs are particularly interesting in this regard, as they evidently do have the capacity, in some sense, for language. However, it's likely that LMs do not use language in the same way that humans do. As Shanahan (2024) writes: "Humans learn language through embodied interaction with other language users in a shared world, whereas a large language model is a disembodied computational entity..." So we may doubt that the way in which LMs use language is indicative of ToM. What we might really care about is whether LMs can engage in genuine, ToM-dependent, *communicative interaction* (Frankish, 2024).

Theories of *communication* typically rely on how we use language to act, and what we *mean* when we use it (Green, 2021; Speaks, 2024). Grice's influential theory of communicative meaning defines a person's *meaning something* through an utterance in terms of the speaker's intentions and the audience's *recognition* of those intentions. Specifically, Grice requires a *third order intention:* the utterer (U) must *intend* that the audience (A) *recognises* that U *intends* that A produces a response (such as a verbal reply). So higher-order ToM is a pre-condition for linguistic communication (Dennett, 1988).

Whilst it may be premature to commit to any particular theory of language use, AI persons should have sufficient ToM to interact with other agents in a full sense, including to cooperate and communicate, or for malicious purposes, e.g., to manipulate or deceive them. Hence, our second condition is as follows. Here, because linguistic communication requires ToM, 2.1 is taken to be a pre-requisite for 2.2.

**Condition 2: Theory-of-Mind and Language.**

> 1. An AI system has *theory-of-mind* to the extent that it has higher-order intentional states, such as beliefs about the beliefs of other agents.

> 2. AI persons should be able to use their ToM to interact and communicate using language.

A number of recent works evaluate contemporary LMs on ToM tasks from psychology, such as understanding false beliefs, interpreting indirect requests, and recognising irony (van Duijn et al., 2023; Strachan et al., 2024; Ullman, 2023). Results are mixed: SOTA LMs sometimes outperforming humans (Strachan et al., 2024; van Duijn et al., 2023), but performance appears highly sensitive to prompting and training details (van Duijn et al., 2023; Ullman, 2023). van Duijn et al. find that fine-tuning LMs to follow instructions increases performance, hypothesising that this is because it "[rewards] cooperative communication that takes into account interlocutor and context".

**Self-Awareness.** Self-awareness plays a central role in theories of personhood (Frankfurt, 2018; Dennett, 1988; Smith, 2024). For instance, Locke (1847) characterises a person as: "a thinking intelligent Being, that has reason and reflection, and can *consider itself as itself*, the same thinking thing in different times and places." But what does it mean, exactly, to be self-aware?

First, persons can know things about themselves in just the same way as they know other empirical facts. For instance, by reading a textbook on human anatomy I can learn things about myself. Similarly, an LM may "know" facts about itself, such as its architectural details, if such facts were

included in its training data. In this sense, someone may have knowledge about themselves without additionally knowing that it applies to them.

Laine et al. present a benchmark for evaluating whether LMs know facts about themselves, including which entity it is, and what detailed properties it has (e.g. its architecture, training cutoff date). Contemporary models perform significantly worse than human baselines, but better than chance, and, similar to ToM tasks, fine-tuning models to interact with humans improves performance.

Second, some of my knowledge is *self-locating*, meaning that it tells me something about my position in the world (Egan and Titelbaum, 2022) – as when Perry sees that someone in a shop is leaving a trail of sugar, and then comes to know that it is *he himself* that is making the mess (Perry, 1979). Self-locating knowledge has behavioural implications which may make it amenable to evaluation in AI systems (Berglund et al., 2023). For instance, an AI system may know that certain systems should send regular updates to users, but may not know that *it* is such a system, and so may not send the updates.

Third, we, as human persons, have what philosopher's call "self-knowledge" – knowledge of our mental states (Gertler, 2024). As humans, we have awareness of our mental states, such as our beliefs and desires, and we acquire self-knowledge via introspection (Schwitzgebel, 2024b). We have a certain special access, unavailable to other agents, to what goes on in our mind.

Anon. (2024) define introspection in the context of LMs as "a source of knowledge for an LLM about itself that does not rely on information in its training data." They provide evidence that contemporary LMs predict their own behaviour using "internal information" such as "simulating its own behavior". Furthermore, LMs "know what they know", i.e., they can predict which questions they will be able to answer correctly (Kadavath et al., 2022), and "know what they don't know": they can identify unanswerable questions (Yin et al., 2023). Laine et al. measure whether LMs can "obtain knowledge of itself via direct access to its representations", for example, by determining how many tokens are used to represent part of its input (this information is dependent its architecture and is unlikely to be contained in training data). Interestingly, Treutlein et al. find that, when trained on input-output pairs of an unknown function $f$, LMs can describe $f$ in natural language without in-context examples. For example, in one experiment, they fine-tune an LM on a corpus consisting only of distances between an unknown city and other known cities. Remarkably, the LM can verbalize that the unknown city is Paris and use this fact to answer downstream questions zero-shot. These results seem to suggest that contemporary LMs have some ability to introspect on their internal algorithmic processes.

Fourth, we have the ability to *self-reflect*: to take a more objective stance towards our picture of the world, our beliefs and values, and the process by which we came to have them, and, upon this reflection, to change our views (Nagel, 1989). Self-reflection plays a central role in theories of personal-autonomy (Buss and Westlund, 2018), i.e., the capacity to determine one's own reasons and actions, which, in turn, is an important condition for personhood (Frankfurt, 2018; Dennett, 1988). More specifically, Frankfurt claims that *second-order volitions*, i.e., preferences about our preferences, or desires about our desires, are "essential to being a person". Importantly, self-reflection enables a person to "induce oneself to change" (Dennett, 1988). To our knowledge, no work has been done to evaluate this form of self-reflection in AI systems, and no contemporary system could plausibly be described as engaging in it. Hence, we decompose self-awareness as follows.

**Condition 3: Self-awareness.** AI persons should be *self-aware*, including having a capacity for:

1. *Knowledge about themselves:* e.g., knowing facts such as its architectural details;

2. *Self-location:* knowing that certain facts apply to *itself* and acting accordingly;

3. *Introspection:* an ability to learn about itself via "internal information" – i.e., without relying on information in its training or context;

4. *Self-reflection:* an ability to take an objective stance towards itself *as an agent in the world* (Nagel, 1989), to evaluate itself, and to induce itself to change (Buss and Westlund, 2018).

**Conclusion.** We present three conditions which, we argue, an AI system needs to satisfy to be considered a person: agency, theory-of-mind, and self-awareness. We claim that no contemporary AI system sufficiently satisfies every condition.Taking seriously the possibility of advanced, misaligned AI systems, Russell is led to ask, "How can humans maintain *control* over AI — forever?" (Russell, 2023). However, the framing of control may be untenable if the AI systems we create are *persons* in their own right. Moreover, unjust repression often leads to revolution (Goldstone, 2001). In this paper, we aim to make progress toward a world in which humans harmoniously coexist with our future creations.

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
