# OpenReview forum: "Towards a Theory of AI Personhood"
_NeurIPS.cc/2024/Workshop/SafeGenAi — SafeGenAi Poster_

### Official Review · Reviewer_jFyJ · 2024-10-08
**Ambitious paper. Interesting thoughts. Unclear on fit? Would benefit the workshop if better structured.**

**Rating:** 6
**Confidence:** 4

**Review:**

This paper proposes conditions for AI personhood. While many other people are cited, there are no other criteria for personhood created by others visible in this paper. Including those would help to contextualize the propositions in this paper as well as help make the propositions feel more justified.
Overall I greatly appreciated the author took the time to write a paper on this topic. I completely agree that the conversations about how to use and control AI completely overlook the potential personhood of AI.
On specific comment I had was that the motivation for grouping language and Theory of Mind together did not well motivated. The links made sense but there were not compelling to me at least. Some people can't speak but that does not remove their personhood.
This reads more a lot more like an ethics paper than what I would expect for a machine learning workshop, making me unsure of whether this paper is a good fit. It does a good job citing from cognitive science and machine learning literatures, but the reasoning does not feel as concrete as scientists tend to prefer. That being said, I personally believe there should be more intermingling of disciplines and the current tendency for only like-minded people to come together in their own workshops seems to sustain all the issues of silos that academia complains about. I think any effort that the author could put into making the paper better structured and would go a long way in helping scientists engage meaningfully with the propositions in this paper.

---

### Official Review · Reviewer_vppa · 2024-10-09
**AI Personhood: Valuable Discussion - Broadly considered SafeGenAI**

**Rating:** 6
**Confidence:** 4

**Review:**

Review Overview:\
This paper discusses the conditions for AI to be considered a person, provides details, describes past works that measure these conditions, and concludes that current AI systems should not be considered persons. I believe the work lacks a discussion of "safety" in regards to personhood, so this may not be the best venue; however, they provide a fascinating discussion that is worth sharing with the community.\
This review follows the sections of the paper. Strengths (+) and weaknesses (-) are noted for each section.

Introduction:\
\+ Strong intro providing relevant overview and background.\
\- The third paragraph mentions human-like properties, and the last paragraph states 3 that constitute a person. The paper can be strengthened by providing reasons why all the others mentioned in the third paragraph of the intro (belief, intent, consciousness, etc.) are not necessary for personhood or are encompassed in these 3.

Conditions on Personhood:\
\+ Well organized, easy to follow, good flow.\
\- When discussing RL, it might be that the agent's "intentions, beliefs, desires and goals" as mentioned in lines 54 and 82 are actually given and structured by the human who programmed that system; they are not inherently the agent's. Apart from the agent's ability to describe and generalize, it might be beneficial to include some notion of the agent owning/creating its own beliefs and goals in Condition 1. I think the authors may be hinting at this in lines 73-74 but it could be explained more. Or this might be a part of self-awareness 3. "Self-knowledge."\
\- The authors should answer their posed questions in lines 87 and 95 and claim that AI doesn't have agency. If the main claim is "no contemporary solution poses these three conditions of personhood," I believe they should end this section with that conclusion. If the aim of this work is to be an open-ended discussion without taking a stance, they shouldn't introduce that claim in the intro/abstract/conclusion.\
\+ Lines 117-118 provide a good explicit statement to support the authors' claims that LMs do not have ToM.

Conclusion:\
\- Line 200 "unjust repression often leads to revolution" hints at the consequences of considering AI as a person. I think there is more room for explicit discussion of the potential positive and negative consequences of AI fulfilling the 3 conditions for personhood outlined in this work. I think this would wrap well in the theme of "Safe Generative AI."\
\- Also, a succinct description of what future work needs to be done in evaluating AI as a person or what capabilities should be developed from a technical perspective to make AI a person would be beneficial.